# Actions of Esomeprazole on the Maternal Vasculature in Lean and Obese Pregnant Mice with Impaired Nitric Oxide Synthesis: A Model of Preeclampsia

**DOI:** 10.3390/ijms23158185

**Published:** 2022-07-25

**Authors:** Natasha de Alwis, Natalie K. Binder, Yeukai T. M. Mangwiro, Sally Beard, Natasha Pritchard, Elif Kadife, Bianca R. Fato, Emerson Keenan, Fiona C. Brownfoot, Tu’uhevaha J. Kaitu’u-Lino, Natalie J. Hannan

**Affiliations:** 1Therapeutics Discovery & Vascular Function Group, Department of Obstetrics and Gynaecology, The University of Melbourne, Mercy Hospital for Women, Heidelberg, VIC 3084, Australia; natasha.dealwis@unimelb.edu.au (N.d.A.); nkbinder@unimelb.edu.au (N.K.B.); yeukaimangwiro@gmail.com (Y.T.M.M.); sally.beard@unimelb.edu.au (S.B.); bfato@student.unimelb.edu.au (B.R.F.); 2Mercy Perinatal, Mercy Hospital for Women, Heidelberg, VIC 3084, Australia; natasha.pritchard@unimelb.edu.au (N.P.); elif.kadife@unimelb.edu.au (E.K.); emerson.keenan@unimelb.edu.au (E.K.); fiona.brownfoot@unimelb.edu.au (F.C.B.); t.klino@unimelb.edu.au (T.J.K.-L.); 3Obstetrics Diagnostics and Therapeutics Group, Department of Obstetrics and Gynaecology, The University of Melbourne, Mercy Hospital for Women, Heidelberg, VIC 3084, Australia; 4Diagnostics Discovery and Reverse Translation in Pregnancy, The University of Melbourne, Mercy Hospital for Women, Heidelberg, VIC 3084, Australia

**Keywords:** preeclampsia, esomeprazole, nitric oxide, obesity, endothelium

## Abstract

Preeclampsia is a devastating, multisystem disorder of pregnancy. It has no cure except delivery, which if premature can impart significant neonatal morbidity. Efforts to repurpose pregnancy-safe therapeutics for the treatment of preeclampsia have led to the assessment of the proton pump inhibitor, esomeprazole. Preclinically, esomeprazole reduced placental secretion of anti-angiogenic sFlt-1, improved endothelial dysfunction, promoted vasorelaxation, and reduced maternal hypertension in a mouse model. Our understanding of the precise mechanisms through which esomeprazole works to reduce endothelial dysfunction and enhance vasoreactivity is limited. Evidence from earlier studies suggested esomeprazole might work via the nitric oxide pathway, upregulating endothelial nitric oxide synthase (eNOS). Here, we investigated the effect of esomeprazole in a mouse model of L-NAME-induced hypertension (decreased eNOS activity). We further antagonised the model by addition of diet-induced obesity, which is relevant to both preeclampsia and the nitric oxide pathway. Esomeprazole did not decrease blood pressure in this model, nor were there any alterations in vasoreactivity or changes in foetal outcomes in lean mice. We observed similar findings in the obese mouse cohort, except esomeprazole treatment enhanced ex vivo acetylcholine-induced vasorelaxation. As acetylcholine induces nitric oxide production, these findings hint at a function for esomeprazole in the nitric oxide pathway.

## 1. Introduction

Preeclampsia is a devastating, multisystem disorder of pregnancy, affecting between 3–5% of pregnancies [1,2]. Preeclampsia typically begins with the inadequate remodelling of the uterine vasculature by the placenta in early pregnancy [3,4]. In later pregnancy, the subsequent placental malperfusion leads to increased release of soluble factors (including the anti-angiogenic factors soluble fms-like tyrosine kinase and soluble endoglin), culminating in maternal endothelial dysfunction and vascular inflammation [2,4]. This brings about a varying clinical phenotype that can include hypertension, proteinuria, foetal growth restriction and end organ damage to the maternal brain, liver, kidneys, and lungs [2,5]. Unfortunately, once a diagnosis of preeclampsia has been made, there are no curative treatments other than delivery of the foetus, which can cause significant morbidity if preterm [6].

Proton pump inhibitors, such as esomeprazole, are drugs with established pregnancy safety profiles [7,8] that have demonstrated potential as a preeclampsia therapy in pre-clinical studies [9]. They may partially act through the modification of the nitric oxide pathway [9]. Nitric oxide is a potent smooth muscle dilator, and impairment in nitric oxide signalling is likely to play a significant role in the pathophysiological endothelial dysfunction in preeclampsia and potentially its severity [10]. Soluble fms-like tyrosine kinase (sFlt)-1, a key anti-angiogenic factor increased in the maternal circulation in preeclampsia [11], may also impair the phosphorylation of endothelial nitric oxide synthase (eNOS), an enzyme that produces nitric oxide [12]. In earlier studies conducted by our group, human umbilical vein endothelial cells (HUVECs) treated with esomeprazole had significantly increased the phosphorylation of eNOS [9]. Treatment with esomeprazole in a mouse model of preeclampsia, induced by the over-expression of sFlt-1 in the placenta, was able to reduce the hypertensive phenotype; however, treatment of chronically hypertensive pregnant mice lacking eNOS did not have the same effect [9]. Overall, this pre-clinical data suggested that esomeprazole may be a potential treatment for preeclampsia, working through the nitric oxide pathway.

Obesity is associated with an increased risk of developing preeclampsia [13]. Increasing numbers of overweight or obese women are entering pregnancy worldwide [14,15,16,17]. This may be due to both impaired early placental development predisposing to release of pathogenic circulating soluble factors [18], as well as underlying maternal endothelial dysfunction making the clinical manifestations more pronounced [19,20]. Nitric oxide bioavailability is reduced in obese individuals [21], and pharmacological studies have indicated that nitric oxide plays a role in the regulation of food intake, energy expenditure, and both glucose and lipid metabolism [22]. Therefore, it is plausible that a preeclampsia treatment that works through the nitric oxide pathway may have additional positive effects in an obese cohort.

Within this context, the current study had two main aims. Firstly, we aimed to assess the effect of esomeprazole in a mouse model of preeclampsia whereby the pathology was instigated through targeting the nitric oxide pathway, specifically by blocking nitric oxide synthase. This was achieved through the use of N(ω)-nitro-L-arginine methyl ester (L-NAME). Secondly, we aimed to assess the additional effect of obesity on this pathway, by assessing the effect of esomeprazole in the same mouse model of preeclampsia fed a high-fat diet with associated excessive weight gain.

## 2. Results

### 2.1. Esomeprazole Does Not Alter Blood Pressure in Lean Mice

Hypertension is a key characteristic of preeclampsia. Therefore, we first assessed whether treatment with esomeprazole could lower the elevated blood pressure induced by L-NAME in our model of preeclampsia [23]. Mice treated with L-NAME that also received esomeprazole did not have altered body weight, nor fat pad weight compared to control (Appendix A). Esomeprazole administration did not lower systolic (Appendix A), diastolic (Appendix A), or mean arterial blood pressure (Figure 1A,B) at either gestational day (D)14.5 or D17.5 of pregnancy, two timepoints at which we have previously demonstrated significantly increased blood pressure with L-NAME treatment.

### 2.2. Esomeprazole Treatment of Lean L-NAME Mice Does Not Alter Vascular Reactivity

Preeclampsia is associated with maternal systemic vasoconstriction, leading to hypertension and impairing blood supply to the major organs. Our previous studies demonstrated that esomeprazole directly induced the vasorelaxation of human omental arteries through endothelium mediated actions [9]. Hence, we assessed whether esomeprazole administration could alter the vascular reactivity of the mesenteric arteries of the mice receiving L-NAME in our model of preeclampsia. Esomeprazole administration in vivo throughout pregnancy did not affect ex vivo maternal mesenteric artery vasorelaxation to acetylcholine (Figure 2A), nor vasoconstriction to phenylephrine (Figure 2B). The analysis of the LogEC50 (shift of curve), area under the curve (total response over experiment) and maximum responses were also unchanged from control (Appendix A).

### 2.3. Esomeprazole Treatment of Lean Mice Receiving L-NAME Does Not Alter Circulating Constrictor, Anti-Angiogenic, or Inflammatory Factors

Preeclampsia is associated with significantly increased levels of vasoconstrictor endothelin-1 (ET-1), anti-angiogenic factor sFlt-1, and inflammatory marker C-reactive protein (CRP) in the maternal circulation [24,25], which are thought to contribute to systemic vascular dysfunction. We have also previously demonstrated that these factors are elevated in the maternal circulation of mice given L-NAME [23]. Here, we assessed whether esomeprazole treatment could reduce the levels of these circulating factors, hence possibly mitigating their action that drives vascular injury and dysfunction. Esomeprazole treatment did not alter the levels of ET-1, sFlt-1, or CRP in lean mice administered L-NAME in pregnancy (Figure 3A–C, respectively).

### 2.4. Esomeprazole Treatment Did Not Improve Fetal or Placental Growth in Lean Mice Receiving L-NAME

Preeclampsia is often associated with impaired placental development (especially early onset preeclampsia), which can have negative impacts on foetal growth. The L-NAME mouse model of preeclampsia also demonstrates impaired foetal and placental development [23]. Hence, we assessed whether esomeprazole could reduce the impact of blocking NO and improve placental and foetal growth. There was no evidence of improvement; foetuses of lean (normal weight) mice administered esomeprazole alongside L-NAME in pregnancy did not have altered weight or crown-to-rump length, in addition there was no difference in placental weight or the foetal:placental weight ratio (Figure 4). Further analysis based on foetal sex did not produce significantly different results (Appendix A). Esomeprazole treatment did not alter litter size and there was no significant difference in the number of female or male foetuses between the groups (data not shown).

### 2.5. Esomeprazole Treatment Did Not Alter Expression of Genes Involved in Kidney Function and Did Not Improve Histopathological Changes in the Kidney or Liver

Preeclampsia can cause major organ damage, including kidney and liver dysfunction. Histopathological analysis demonstrated no difference in kidney glomeruli damage, swelling, increased cellularity or the narrowing of the Bowman’s space between esomeprazole and vehicle control groups (Figure 5A,B). We also assessed whether esomeprazole treatment might alter the expression of genes associated with kidney function in kidneys collected from this model. The expression of *Hsd11b2*, *Nox4*, *Fn1*, *Sgk1*, and *Sccn1a* in kidneys from lean mice administered L-NAME and esomeprazole was not significantly different from that of the control mice (no esomeprazole; Appendix A). Liver histopathology showed no differences in micro/macro vesicular steatosis, nuclear displacement, or fibrosis between the treated and control groups (Figure 5C,D).

### 2.6. Esomeprazole Did Not Significantly Alter Blood Pressure of Obese Mice Administered L-NAME

Obese individuals are at an increased risk of preeclampsia [26], additionally obesity can alter vascular function [27]. Hence, we were interested to know whether esomeprazole, which in preclinical studies has shown direct action on the maternal vasculature [9], had positive actions on vascular function in obese mice on a high fat diet. Mice on a high fat diet and administered esomeprazole alongside L-NAME did not have significantly altered body weight compared to controls (Appendix A). However, their fat pad weights tended to be higher than controls (though this was not significantly different; Appendix A).

Esomeprazole treatment has previously been shown to reduce the hypertensive phenotype in another mouse model of preeclampsia (placental sFlt-1 overexpression) [9]. However as reported above, esomeprazole did not reduce blood pressure in lean (normal weight) mice administered L-NAME to induce the preeclamptic phenotype. We assessed whether obese mice may respond differently; however, mean blood pressure was not significantly altered with esomeprazole administration in obese mice receiving L-NAME compared to those that were not administered esomeprazole (vehicle control) at either D14.5 (Figure 6A) or D17.5 (Figure 6B) of pregnancy. Systolic and diastolic blood pressures were also unchanged at both time points (Appendix A).

### 2.7. Vasodilation to Acetylcholine Is Enhanced in Mesenteric Arteries Collected from Obese Mice Receiving L-NAME and Treated with Esomeprazole

Esomeprazole directly induced the vasodilation of human omental arteries from preeclamptic women [9]. However, here we demonstrate that mesenteric arteries collected from lean mice receiving L-NAME did not have an altered vascular response to acetylcholine or phenylephrine when esomeprazole was administered throughout pregnancy (in vivo). Here, we assessed whether esomeprazole action may differ on mesenteric arteries collected from obese pregnant mice. Mesenteric arteries collected from the obese L-NAME mice administered esomeprazole had significantly increased vasodilation to acetylcholine at two doses (10^−7^ M *p* = 0.0372, 10^−6.5^ M *p* = 0.003; Figure 7A), compared to arteries from vehicle control treated mice. LogEC50, area under the curve and maximum relaxation were not altered in the arteries from the obese mice given esomeprazole compared to the controls (Appendix A). Mesenteric arteries did not have altered constriction to phenylephrine compared to controls (Figure 7B), nor was LogEC50, area under the curve, or maximum constriction altered (Appendix A).

### 2.8. Esomeprazole Did Not Alter Levels of Circulating ET-1, sFlt-1 or CRP in Obese Mice Administered L-NAME

We next aimed to establish whether factors in the maternal circulation that contribute to endothelial dysfunction may be altered with esomeprazole treatment in obese mice. Esomeprazole treatment in these mice did not alter the maternal serum levels of ET-1, sFlt-1, or CRP (Figure 8A–C, respectively).

### 2.9. Esomeprazole Did Not Improve Fetal Size but Reduced Placental Weight in Obese Mice Receiving L-NAME

We next investigated whether esomeprazole treatment altered foetal and placental development in obese L-NAME mice. Esomeprazole did not improve foetal weight in obese dams receiving L-NAME (Figure 9A) but unexpectedly reduced placental weight (*p* = 0.0404; Figure 9B). However, the foetal to placental weight ratio, an indicator of placental function, was not significantly altered (Figure 9C). Esomeprazole did not significantly alter crown to rump length (Figure 9D). Further analysis based on foetal sex did not produce significantly different results (Appendix A). Similar to the lean dams, the obese mice had no difference in litter size or in the number of male/female foetuses (data not shown).

### 2.10. Esomeprazole Does Not Alter Expression of Genes Involved in Kidney Function, nor Histopathology of the Kidney or Liver in Obese Mice Administered L-NAME

We next assessed whether esomeprazole may be able to mitigate kidney dysfunction associated with a preeclampsia phenotype in obese mice. Obesity in these L-NAME treated mice was associated with pathological changes in the kidney, including enlarged glomeruli, the narrowing of the Bowman’s capsule, and immune infiltrations, which were not improved with esomeprazole treatment (Figure 10A,B). Similarly, esomeprazole did not alter kidney expression of *Hsd11b2*, *Nox4*, *Fn1*, *Sgk1*, and *Sccn1a* in comparison to control (Appendix A). Obesity in these L-NAME treated mice was also associated with pathological changes in the liver, including steatosis with nuclear displacement, which also was not improved with esomeprazole treatment (Figure 10C,D).

## 3. Discussion

Despite increasing research efforts, delivery remains the only definitive treatment for preeclampsia. While this effectively cures the mother, delivery at preterm gestations can confer significant morbidity to the baby. We have previously demonstrated that the proton pump inhibitor, esomeprazole, ablates several key characteristics of the pathophysiology of preeclampsia in pre-clinical models. Here we aimed to further investigate the potential action of esomeprazole in the nitric oxide pathway, in the context of preeclampsia. Employing N(ω)-nitro-L-arginine methyl ester (L-NAME) to inhibit nitric oxide production in pregnant mice, we showed that esomeprazole was unable to decrease blood pressure; improve foetal or placental growth; or quench circulating anti-angiogenic, vasoconstrictor and inflammatory factors in both mice on a normal and high fat diet. However, we found that systemic arteries from obese dams treated with esomeprazole had enhanced ex vivo vasorelaxation to acetylcholine.

Nitric oxide is a potent vasodilator, synthesised from L-arginine by endothelial nitric oxide synthase (eNOS). Nitric oxide activates soluble guanylyl cyclase in nearby vascular smooth muscle cells, which in turn converts GTP to cGMP, activating cGMP-dependent protein kinase, ultimately resulting in vasodilation. Numerous studies have reported dysregulated plasma nitrate/nitrite, a surrogate marker of in vivo nitric oxide production, with preeclampsia [28,29,30,31,32,33,34,35,36]. Previously, we presented evidence suggesting that esomeprazole may be acting through the nitric oxide pathway [9], where esomeprazole enhanced the phosphorylation of eNOS in human umbilical vein endothelial cells [9]. Furthermore, despite esomeprazole potently reducing blood pressure in our sFlt-1 overexpression mouse model of preeclampsia, esomeprazole was unable to significantly reduce blood pressure in the chronic hypertensive eNOS knockout pregnant mouse [9], further suggesting that esomeprazole may be working through the nitric oxide pathway.

Unlike the eNOS knockout mouse where eNOS is completely absent, we used N(ω)-nitro-L-arginine methyl ester (L-NAME) to decrease nitric oxide production by inhibiting the enzyme NOS in a mouse model of preeclampsia [23]. L-NAME is hydrolysed by cellular esterases to the bioactive NG-nitro-L-arginine in the vascular endothelium, where it inhibits NOS activity by binding to the enzyme in place of its normal substrate, L-arginine [37]. The L-NAME inhibition of NOS is reversible, and the extent of inhibition correlates to the dose of L-NAME given and time since administration [37]. If esomeprazole is working to increase the phosphorylation of eNOS, and thus eNOS activity, we expect it to reduce the L-NAME induced inhibition, promote vasorelaxation and mitigate the characteristics of preeclampsia that are synonymous with this animal model.

Our L-NAME mouse model of preeclampsia demonstrates increased blood pressure, impaired foetal and placental growth, and increased circulating levels of ET-1, sFlt-1, and CRP [23]. However, in lean (normal weight) dams, esomeprazole was unable to mitigate this increased blood pressure (much like in the eNOS knockout mouse [9]), nor any other maternal or foetal/placental parameters assessed. This suggested that esomeprazole could not overcome the L-NAME-induced inhibition of nitric oxide. Thus, suggesting its action to reduce blood pressure as seen in the sFlt-1 overexpression model [9] is likely due to action through this pathway.

We further challenged the model by the addition of diet-induced obesity. As well as there being a strong association between preeclampsia and obesity [13], obesity is also associated with decreased nitric oxide bioavailability [21]. L-arginine is the only substrate for endothelial nitric oxide formation, and obesity compromises the transport of extracellular L-arginine via the cationic amino acid transporter [38]. Using this model of L-NAME and obesity, we thus aimed to insult the synthesis of nitric oxide at two points, substrate transport and enzyme activity.

Similar to the lean mice, esomeprazole did not reduce blood pressure in the obese L-NAME treated mice, nor was it able to elicit a change in ex vivo vasoconstriction, circulating levels of ET-1, sFlt-1 and CRP, the foetal, or kidney/liver parameters. In contrast, however, esomeprazole enhanced ex vivo vascular relaxation in response to acetylcholine in the obese cohort. Acetylcholine induces vasorelaxation through the nitric oxide pathway by upregulating nitric oxide production [39,40]. The vasorelaxant effect of acetylcholine is blunted in the presence of L-NAME [41,42,43]. Taken together, this suggests that esomeprazole may act through the nitric oxide pathway to enhance vasodilation. This might occur in the obese cohort of mice but not the lean (normal weight) cohort due to different drug pharmacokinetics with obesity. In humans, adipose tissue in obese women has increased NOS expression (both endothelial and inducible) [44], and L-NAME elicits different effects in obese people compared to lean controls [45]. Assessing vascular responses with other dilating and constricting agonists may further validate the mechanisms by which esomeprazole acts on the vasculature.

Esomeprazole treatment caused a significant decrease in placental weight in the obese animals, comparable to placental weights from lean mice receiving L-NAME. Placental weight correlates positively with maternal weight [46]. However increased size does not confer a better-quality placenta, with reports showing that obesity is associated with non-branching placental angiogenesis [47] and decreased blood flow in hypertensive and preeclamptic pregnancies [48]. Whether this placental change relates to the nitric oxide pathway is unknown. The further investigation of esomeprazole in obese models without L-NAME treatment would yield useful insight into these finding.

Human clinical trials have shown mixed results in the use of esomeprazole to treat preeclampsia. One study demonstrated that there was no prolongation of gestation, improvement of maternal or neonatal outcomes, or decrease in circulating sFlt-1 and markers of endothelial dysfunction in patients with pregnancies complicated by preterm preeclampsia [49]. A second study has since claimed the prolongation of gestation and increased foetal weight in patients with preterm preeclampsia [50]. By improving our understanding of the mechanisms behind esomeprazole action, we can identify the limitations, and design better clinical trials. This includes suggesting more effective doses of esomeprazole, identifying the populations that would most benefit from using esomeprazole, or recommending alternative routes of administration for improved clinical effect. Interestingly, the oral use of proton pump inhibitors, such as esomeprazole, has been associated with decreased bioavailable nitric oxide [51,52,53]. Separate to the L-arginine/NOS pathway of nitric oxide production, a significant proportion of nitric oxide is derived from dietary nitrates and nitrites [54]. Proton pump inhibitors work to increase stomach pH, which decreases the efficiency of dietary nitric oxide production [51,52,53]. Clinical trials could avoid this deleterious effect on dietary nitric oxide production by using intraperitoneal injection, as we did in this current study. 

This study aimed to interrogate the potential impact of esomeprazole in the nitric oxide pathway in the context of preeclampsia both with and without obesity. While we have presented largely negative findings, the identification of increased vasorelaxation with acetylcholine and esomeprazole combined further supports that esomeprazole may play a role in increasing nitric oxide production in the context of preeclampsia. Increased nitric oxide would likely have beneficial vasoactive properties and is important to continue to explore for the treatment of preeclampsia.

## 4. Materials and Methods

### 4.1. Animal Studies

Animal experiments were approved by the Austin Health Animal Ethics Committee (A2018/05596) and followed the National Health and Medical Research Council ethical guidelines for the care and use of animals for scientific purposes. Three-week-old CBA x C57BL/6 (F1) female mice (n = 40) were sourced from the Florey Institute of Neuroscience and Mental Health (The University of Melbourne). Mice were group-housed in conventional open-top cages, on a 12-hour light/dark cycle, with food and water available ad libitum (18–22 °C; 50% relative humidity). At 4 weeks, F1 females were randomly allocated to consume a standard chow diet or a Western style high fat diet (21% fat (40% total energy from fat)) (SF05-31; Specialty Feeds, Glen Forrest, WA, Australia). Mice were also acclimated to the CODA non-invasive blood pressure system (Kent Scientific, Torrington, CT, USA) in a seven-stage process involving exposure to the restraining tube and tail cuff.

From 8 weeks of age, F1 females were mated overnight with stud F1 male mice. Pregnancy was confirmed by the presence of a copulatory plug the following morning; designated as gestational day (D)0.5.

### 4.2. L-NAME Mouse Model of Preeclampsia and Esomeprazole Treatment

L-NAME was used to induce a preeclampsia-like phenotype as previously described [23]. L-NAME (50 mg/kg/day; Sigma-Aldrich, St. Louis, MO, USA) was administered daily from D7.5 to D17.5 of pregnancy (approximately early second trimester to term in human equivalent) via 100 µL subcutaneous injection. At D7.5, mice were randomly allocated to either the treatment group receiving esomeprazole at 10 mg/kg or the control group, receiving the vehicle (water) via daily intraperitoneal injection.

### 4.3. Tissue Collection

Blood pressure was measured on D14.5 and D17.5 of pregnancy. Following blood pressure and maternal body weight measurement on D17.5, mice were anesthetized with 5% isoflurane in oxygen, and cardiac puncture was performed to collect maternal blood. Mice were then culled by cervical dislocation. Blood samples were allowed to coagulate at room temperature before centrifugation to separate the serum fraction, which was snap frozen and stored at −80 °C.

Foetuses and placentas were counted and weighed, foetal crown-to-rump length measured (with digital callipers), and sex of each foetus determined [55]. Maternal kidneys were placed in RNAlater for a minimum of 48 h (as per the manufacturer’s recommendations), then snap frozen and stored at −80 °C until subsequent analysis. Maternal livers and kidneys were fixed in a 10% neutral buffer overnight for tissue histology. The intestinal tract was collected in ice cold PBS for the dissection of mesenteric arteries for vascular studies. Maternal fat pads were dissected and weighed.

### 4.4. Maternal Kidney and Liver Histology

After fixation in 10% neutral buffered formalin overnight, livers and kidneys were washed in PBS and embedded in paraffin for sectioning at 4 uM thickness. The sections were deparaffinized in xylene and rehydrated through descending grades of ethanol prior to haematoxylin and eosin staining. Tissue histology was visualized and captured using a Nikon Eclipse Ci microscope and camera at 100 µm magnification (n = 3 sections/condition).

Regions from each lobe of the liver were dissected and embedded for each animal. For the liver, steatosis features, including nuclear displacement, necrosis, inflammatory infiltration, fibrosis and the extent of liver involvement, were evaluated within each image. For the kidneys the glomeruli size/cellularity, tubule structures, necrosis/apoptosis and immune infiltrate were considered in each image. The figures represent high magnification images of the regions of interest, demonstrating the overall impact of diet and/or treatment.

### 4.5. Vascular Reactivity Studies

Second order mesenteric arteries were carefully dissected from surrounding connective and adipose tissue in Krebs physiological salt solution (NaCl 120 mM, KCl 5 mM, MgSO_4_ 1.2 mM, KH_2_PO_4_ 1.2 mM, NaHCO_3_ 25 mM, D-glucose 11.1 mM, CaCl_2_ 2.5 mM). Dissected arteries (2 mm length) were then mounted on the 620 M Wire Myograph (Danish Myo Technology (DMT), Hinnerup, Denmark) using 25 µm diameter gold-plated tungsten wires (W005230; Goodfellow, Cambridge, UK) and bathed in Krebs solution continuously bubbled with carbogen (95% O_2_, 5% CO_2_) and warmed to 37 °C. The arteries were normalised to 100 mmHg (13.3 kPa) pressure using the DMT normalisation module on LabChart v8.1.21 software (ADInstruments, Sydney, NSW, Australia) with IC1/1C100 = 1. Smooth muscle viability was confirmed using high potassium physiological salt solution (KPSS; NaCl 25 mM, KCl 100 mM, MgSO_4_ 1.2 mM, KH_2_PO_4_ 1.0 mM, NaHCO_3_ 25 mM, D-glucose 11.1 mM, CaCl_2_ 2.5 mM). Endothelial function was assessed by pre-constricting arteries to 50–70% of maximal constriction to KPSS with phenylephrine (Sigma-Aldrich), then relaxed with the endothelial dependent dilator, acetylcholine (Sigma-Aldrich). Greater than 80% relaxation was required for the inclusion of the vessel. Constriction and relaxation dose response curves were then generated using phenylephrine and acetylcholine (10^−9^ to 10^−4.5^ M).

### 4.6. Enzyme Linked Immunosorbent Assay (ELISA)

Concentrations of soluble fms-like tyrosine kinase 1 (sFLT-1), endothelin-1 (ET-1), and C-Reactive Protein (CRP) in maternal serum were measured using the Mouse sVEGFR1/Flt-1 DuoSet ELISA kit (samples diluted 1:100), Mouse Endothelin-1 Quantikine ELISA Kit, and Mouse C-Reactive Protein/CRP Quantikine ELISA Kit (R&D Systems, Minneapolis, MN, USA) respectively, according to the manufacturer’s instructions.

### 4.7. Quantitative Polymerase Chain Reaction (qPCR)

RNA was extracted from kidneys using RNAeasy Total RNA extraction kit (Qiagen, Valencia, CA, USA) and quantified with a Nanodrop 2000 spectrophotometer (ThermoFisher Scientific, Waltham, MA, USA). Extracted RNA was converted to cDNA using the Applied Biosystems^TM^ High-Capacity cDNA Reverse Transcription Kit, as per manufacturer guidelines on the iCycler iQ5 (Bio-Rad, Hercules, CA, USA). Quantitative PCR with Taqman reagents was performed to quantify mRNA expression using primers purchased from Life Technologies (Carlsbad, CA, USA). Primers used were designed for: fibronectin 1 (*Fn1*; Mm01256744_m1), hydroxysteroid 11-beta dehydrogenase 2 (*Hsd11b2*; Mm01251104_m1), NADPH oxidase 4 (*Nox4*; Mm00479246_m1), serum/glucocorticoid regulated kinase 1 (*Sgk1*; Mm00441380_m1), sodium channel epithelial 1 subunit alpha (*Scnn1α*; Mm00803386_m1), and ubiquitin C (*Ubc*; Mm01198158_m1) as the reference gene.

### 4.8. Statistical Analysis

Data were assessed for normal (Gaussian) distribution and the differences between the treatment and control groups were statistically tested with a Mann–Whitney test and a parametrically or unpaired t-test, as appropriate. Foetal and placental size were assessed using a linear mixed-effects model, with a fixed effect for the treatment group and random effect for each litter. *p*-values were calculated for the treatment effect using nested ANOVA. Myograph dose–response curves were produced using a non-linear regression analysis (log[agonist] vs. response—four parameters). The comparison of responses to the agonist were tested for significance using mixed-effects analysis, with Šidák correction for multiple comparisons. *p*-values < 0.05 were considered significantly different. Statistical analysis was performed using GraphPad Prism 8 software (La Jolla, CA, USA).

## Figures and Tables

**Figure 1 ijms-23-08185-f001:**
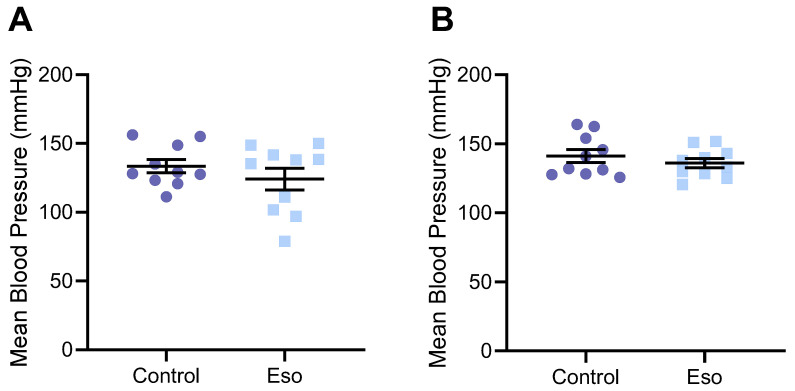
Effect of esomeprazole on blood pressure of lean (normal weight mice on standard chow diet) administered L-NAME in pregnancy. Mean arterial blood pressure was measured (**A**) D14.5 and (**B**) D17.5 of pregnancy via tail cuff plethysmography. Esomeprazole treatment did not alter blood pressure compared to vehicle control (no esomeprazole) at either time point. Results presented as mean ± SEM. n = 10 mice/group.

**Figure 2 ijms-23-08185-f002:**
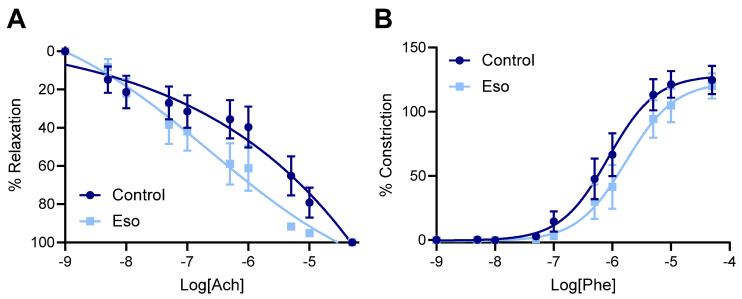
Effect of esomeprazole on the vascular reactivity of mesenteric arteries collected from mice receiving L-NAME through pregnancy. Wire myography was used to assess (**A**) vasorelaxation to acetylcholine (Ach) and (**B**) vasoconstriction to phenylephrine (Phe). Treatment with esomeprazole alongside L-NAME administration in pregnancy did not significantly alter the vascular response to acetylcholine or phenylephrine at any dose assessed. Data points are presented as mean ± SEM, and non-linear regression performed (log(agonist) vs. response–variable slope (four parameters)). n = arteries from 6–8 mice/group.

**Figure 3 ijms-23-08185-f003:**
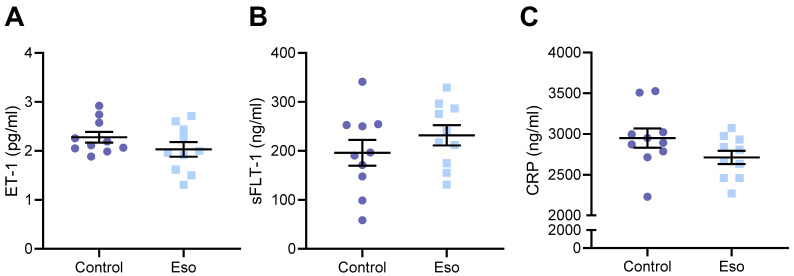
Effect of administration of esomeprazole in lean (normal weight) mice receiving L-NAME on circulating ET-1, sFlt-1, and CRP. Levels of (**A**) ET-1, (**B**) sFlt-1, and (**C**) CRP (measured via ELISA) were not reduced in the serum of mice receiving L-NAME plus esomeprazole in pregnancy, compared to that of the control mice (no esomeprazole). Data presented as mean ± SEM. n = 10 mice/group.

**Figure 4 ijms-23-08185-f004:**
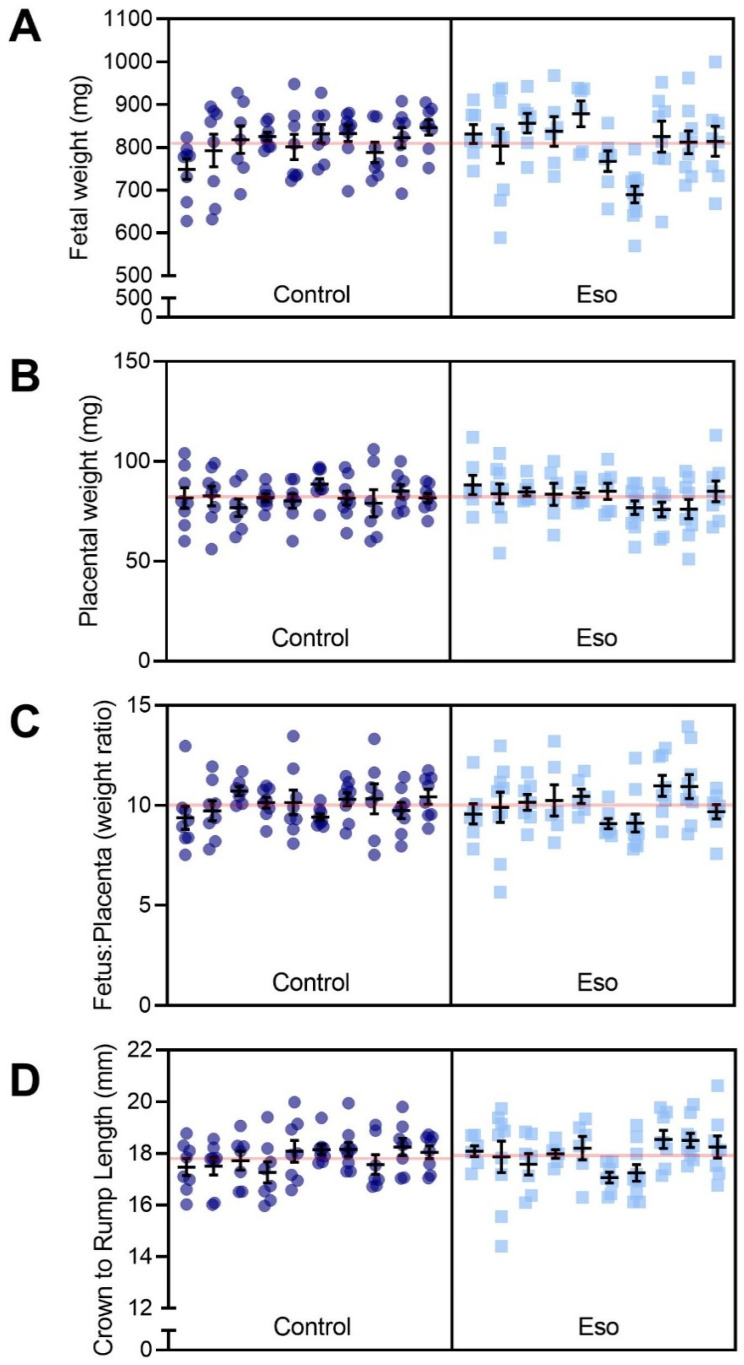
Effect of esomeprazole administration on foetal and placental size in lean (normal weight) mice receiving L-NAME. (**A**) Foetal weight, (**B**) placental weight, (**C**) foetal to placental weight ratio, (**D**) crown to rump length (measured with digital callipers). Esomeprazole administration did not significantly alter either foetal weight or length, nor placental weight. The ratio of foetal to placental weight was also not significantly altered. Results are presented from each dam (n = 10 mice/group) and were assessed using a linear mixed-effects model. Error bars present the mean ± SEM within each litter. The red line across each box represents the mean of each treatment group.

**Figure 5 ijms-23-08185-f005:**
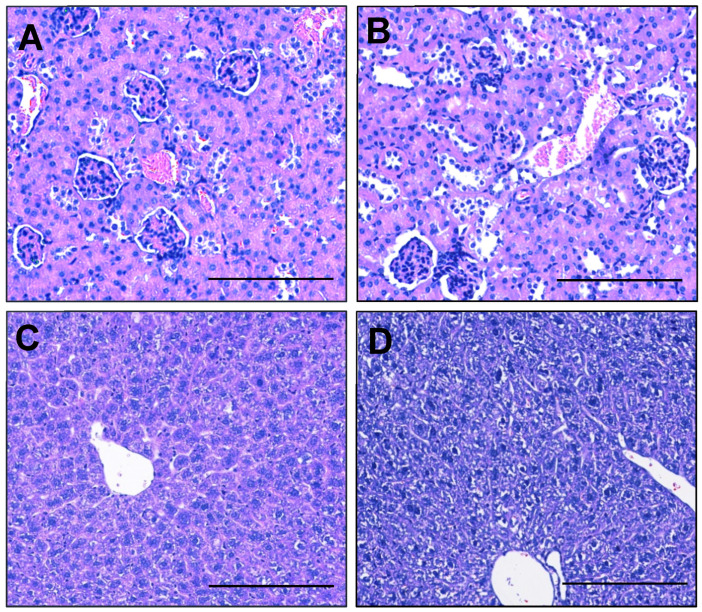
Esomeprazole does not alter kidney or liver morphology of lean pregnant mice receiving L-NAME. Haematoxylin- and eosin-stained sections of kidney did not show differences in glomeruli, Bowmen’s capsule, or immune infiltrations between the (**A**) control and (**B**) esomeprazole group. Assessment of liver showed no difference in micro/macro vesicular steatosis, nuclear displacement, or fibrosis between the (**C**) control and (**D**) esomeprazole group. Scale bar = 100 µm.

**Figure 6 ijms-23-08185-f006:**
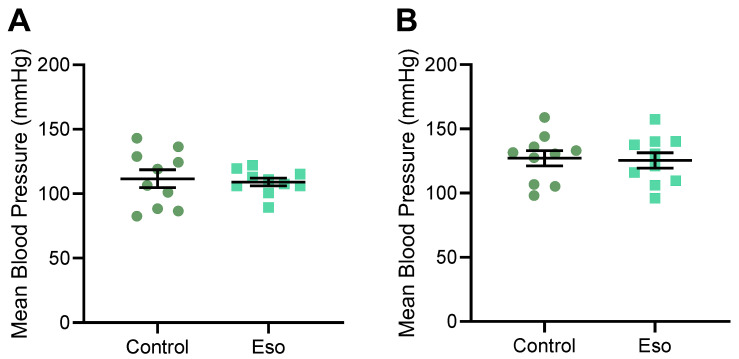
Mean arterial blood pressure of obese mice receiving L-NAME with esomeprazole treatment at (**A**) D14.5 and (**B**) D17.5 of pregnancy. Blood pressure was measured via tail cuff plethysmography. Mean arterial blood pressure was not significantly altered between the obese mice receiving L-NAME, treated with esomeprazole, and the vehicle control treated mice. Results presented as mean ± SEM; n = 10 mice/group.

**Figure 7 ijms-23-08185-f007:**
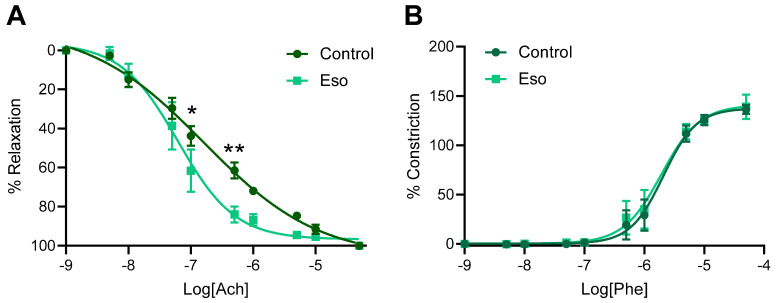
Effect of esomeprazole administration throughout pregnancy on vascular reactivity of mesenteric arteries collected from obese mice receiving L-NAME. Mesenteric artery (**A**) vasodilation to acetylcholine (Ach) and (**B**) constriction to phenylephrine (Phe) was assessed via wire myography (ex vivo). Arteries collected from the obese L-NAME mice given esomeprazole relaxed significantly more in response to 10^−7^ and 10^−6.5^ M doses of acetylcholine (**A**) compared to arteries collected from control (vehicle treated) mice. Arteries did not show altered vasoconstriction in response to phenylephrine compared to the arteries from control mice (**B**). Data points presented as mean ± SEM, and curves fitted with a non-linear regression (log(agonist) vs. response–variable slope (four parameters)). n = arteries from 4–6 mice/group. * *p* < 0.05, ** *p* < 0.01.

**Figure 8 ijms-23-08185-f008:**
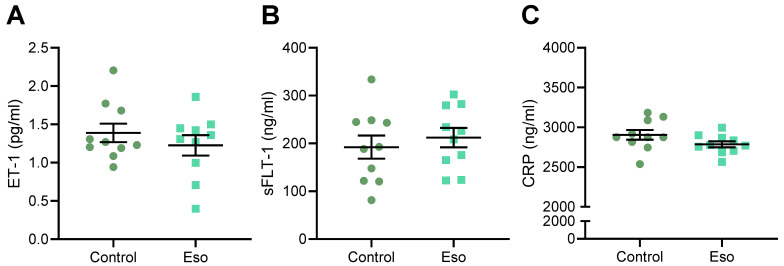
Effect of esomeprazole treatment on circulating (**A**) ET-1, (**B**) sFlt-1, and (**C**) CRP in obese L-NAME mice. Serum levels of these factors were assessed via ELISA. Obese mice receiving L-NAME treated with esomeprazole did not have altered circulating (**A**) ET-1, (**B**) sFlt-1, or (**C**) CRP compared to the controls. Data presented as mean ± SEM; n = 10 mice/group.

**Figure 9 ijms-23-08185-f009:**
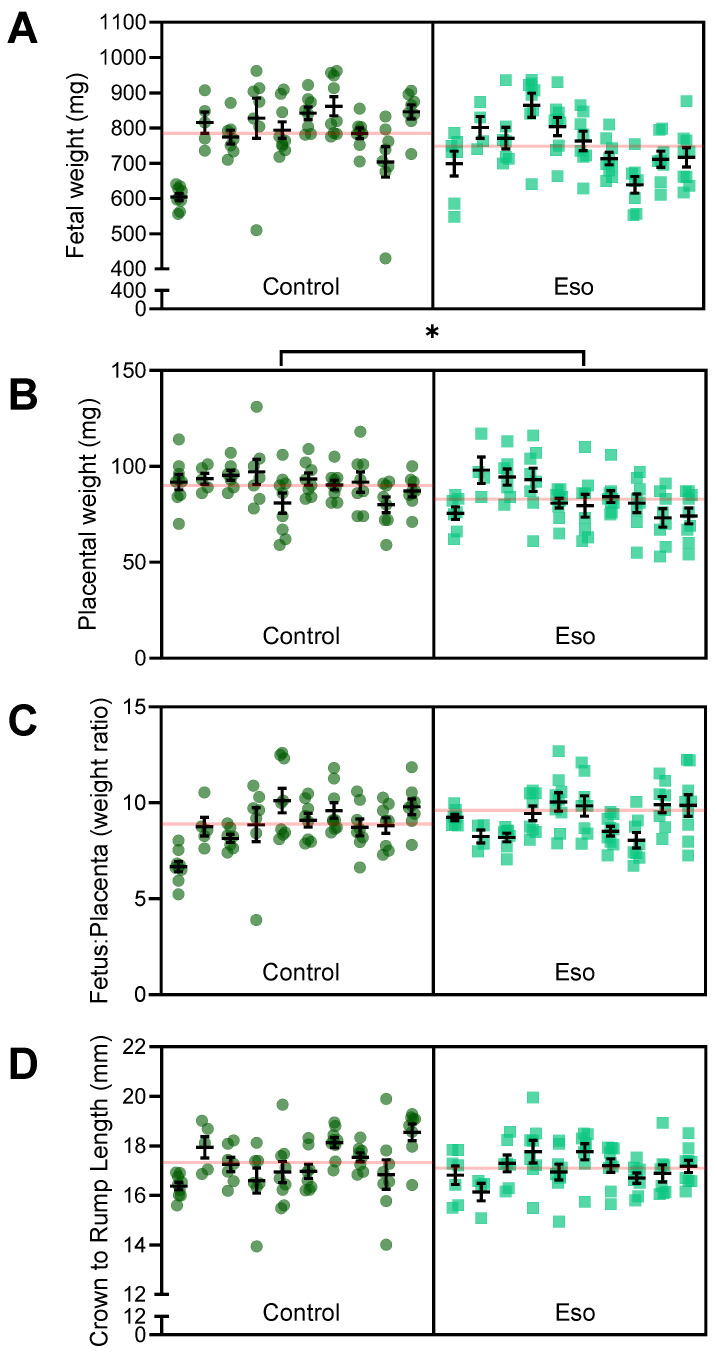
Effect of esomeprazole on foetal and placental weight and size in obese L-NAME mice administered L-NAME. (**A**) Foetal weight, (**B**) placental weight, (**C**) foetal to placental weight ratio, and (**D**) crown to rump length (measured with digital callipers). Esomeprazole administration did not significantly improve foetal weight or length. Placental weight was significantly reduced in the obese L-NAME mice administered esomeprazole. The ratio of foetal to placental weight was not significantly altered. Results are presented from each dam (n = 10 mice/group), mean ± SEM. The red line across each box represents the mean of each group. * *p* < 0.05.

**Figure 10 ijms-23-08185-f010:**
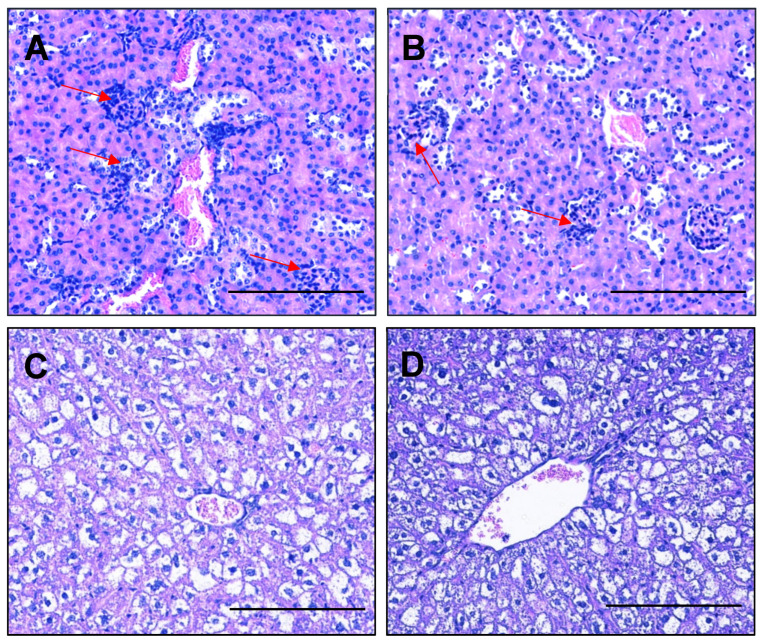
Pathological changes in the kidney and liver of high fat diet L-NAME mice are not improved by esomeprazole treatment. Haematoxylin and eosin-stained kidney sections demonstrated enlarged glomeruli, narrowing of the Bowmen’s capsule space, and immune infiltrations into the glomeruli (red arrows). However, these were found in both the control (**A**) and esomeprazole treated (**B**) mice. Livers from the high fat diet L-NAME mice had steatosis with nuclear displacement (**C**), which was not improved with esomeprazole treatment (**D**). Scale Bar = 100 µm.

## Data Availability

Data available upon reasonable request.

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
