# Peer review of "Actions of Esomeprazole on the Maternal Vasculature in Lean and Obese Pregnant Mice with Impaired Nitric Oxide Synthesis: A Model of Preeclampsia"

_ijms, 2022, doi:10.3390/ijms23158185_

Round 1

Reviewer 1 Report

De Alwis et al. provided a comprehensive research on how esomeprazole involved in NO signaling pathway can impact pregnancy course and preeclampsia in animal models with chronic risk factors. The article provides detail overview of esomeprazole effects on the physiological and histopathological parameters including the assessment of placenta, kidney, liver structures and arterial pressure. 

There are some comments that in my opinion would be helpful to improve the article, which represents a special interest for scientific and medical community, particularly for planning of esomeprazole treatment during pregnancy. 

1. I would recommend to combine together Figures 1 and 6, 2 and 7, 3 and 8, 4 and 9, etc. The result interpretation would be more clear once a reader has a chance to compare the esomeprazole effects in the normal and obese mice in parallel. 

2. I would recommend to present the data of qPCR as 2^-ddCT (Supplementary data). 

3. In severe cases of preeclampsia, not only the blood pressure increases, but the condition is accompanied by swelling, vomiting, visual disturbances, and sometimes coordination in humans. Were any behavioral (for instance, in movement, balance or posture) pathologies discovered throughout the study course and if they were different within the studied groups? Did these symptoms correlate with the blood pressure pattern in the animals?

4. Were there any differences in fetus in the studied models? Ex., weight, sex ration, overall number, etc? 

Minor comments

1. "In later pregnancy, the 40 subsequent placental malperfusion leads to increased release of soluble factors culminating in maternal endothelial dysfunction and vascular inflammation" - please add some examples of the soluble factors 

2. In order to improve the logical connection between paragraphs 2 and 3 of the Introduction I would suggest to re-arrange the beginning of paragraph 3: "Obesity is associated with an increased risk of developing preeclampsia [17]. With the increasing numbers of overweight or obese pregnant women, more attention is drawn to the eclampsia management in obesity worldwide [13-16]. 

3. In the study of vascular reactivity to vasoconstriction, the effect of phenylephrine was investigated. Why was this particular drug chosen? Is there any effect of angiotensin, metoxamine, since these drugs are also used as active agents for vasoconstriction?

4. The statement “Using this model of L-NAME and obesity thus allowed us to insult the synthesis of nitric oxide at two points, substrate transport and enzyme activity” is too strong, from my opinion. The research does not provide enough evidences on altered NO signaling pathway. 

5. Are there any connections with the inflammatory status in control and obese mice treated with the esomeprazole? 

Finally, the current article provides a substantial information on how esomeprazole impact preeclampsia and associated pathologies in normal and obese models. Article can be accepted after the minor revision. 

Author Response

Reviewer 1

De Alwis et al. provided a comprehensive research on how esomeprazole involved in NO signaling pathway can impact pregnancy course and preeclampsia in animal models with chronic risk factors. The article provides detail overview of esomeprazole effects on the physiological and histopathological parameters including the assessment of placenta, kidney, liver structures and arterial pressure.

There are some comments that in my opinion would be helpful to improve the article, which represents a special interest for scientific and medical community, particularly for planning of esomeprazole treatment during pregnancy.

RESPONSE: We thank the reviewer for taking the time to review our manuscript, and for their valuable feedback. We have responded to each of the comments as below.

  1. I would recommend to combine together Figures 1 and 6, 2 and 7, 3 and 8, 4 and 9, etc. The result interpretation would be more clear once a reader has a chance to compare the esomeprazole effects in the normal and obese mice in parallel.

RESPONSE: We thank the reviewer for their suggestion. In this paper, we deliberately separated the lean and obese results as our main focus was the drug effects, rather than the differences between the two cohorts. We do agree that comparing the lean and obese cohorts is of interest, and we will be presenting these differences in future studies.

  1. I would recommend to present the data of qPCR as 2^-ddCT (Supplementary data).

RESPONSE: We thank the reviewer for this suggestion. The data is currently presented in 2^-ddCT, not ddCT – we have corrected this error in the y-axis of each graph. Please see Supplementary Figure S5 and S10.

  1. In severe cases of preeclampsia, not only the blood pressure increases, but the condition is accompanied by swelling, vomiting, visual disturbances, and sometimes coordination in humans. Were any behavioral (for instance, in movement, balance or posture) pathologies discovered throughout the study course and if they were different within the studied groups? Did these symptoms correlate with the blood pressure pattern in the animals?

RESPONSE: We thank the reviewer for their query. Assessing changes in behaviour was out of the scope for this study. We did not study behaviour or movement above the basic observation required for our animal monitoring, with which we did not see any obvious changes.

  1. Were there any differences in fetus in the studied models? Ex., weight, sex ration, overall number, etc?

RESPONSE: Esomeprazole did not alter fetal weight, fetal to placental weight ratio, or fetal crown to rump length in both the lean and obese mice, as presented in Figures 4 and 9. Placental weight was significantly reduced in the esomeprazole group in obese mice (Figure 9). These parameters were not different with fetal sex, as demonstrated in Supplementary Figures S4 and S9.

Esomeprazole treatment also did not affect total litter size, or selectively affect survival of either fetal sex. To clarify this, we have added a statement to the relevant results, as below.

Page 4, lines 148-149: “Esomeprazole treatment did not alter litter size, and was there no significant difference in the number of female or male fetuses between the groups (data not shown).”

Page 9, lines 246-248: “Similar to in the lean dams, the obese mice had no difference in litter size or in the number of male/female fetuses (data not shown).”

Minor comments

  1. "In later pregnancy, the 40 subsequent placental malperfusion leads to increased release of soluble factors culminating in maternal endothelial dysfunction and vascular inflammation" - please add some examples of the soluble factors

RESPONSE: We thank the reviewer for this suggestion. We have added examples of the soluble factors as follows.

Page 1, Lines 41-42: “…leads to increased release of soluble factors (including the anti-angiogenic factors fms-like tyrosine kinase and soluble endoglin), culminating in maternal endothelial dysfunction and vascular inflammation [2,4].”

  1. In order to improve the logical connection between paragraphs 2 and 3 of the Introduction I would suggest to re-arrange the beginning of paragraph 3: "Obesity is associated with an increased risk of developing preeclampsia [17]. With the increasing numbers of overweight or obese pregnant women, more attention is drawn to the eclampsia management in obesity worldwide [13-16].

RESPONSE: We thank the reviewer for their suggestion. We have altered the order of these lines as advised. Please see Page 2, Lines 65-66.

  1. In the study of vascular reactivity to vasoconstriction, the effect of phenylephrine was investigated. Why was this particular drug chosen? Is there any effect of angiotensin, metoxamine, since these drugs are also used as active agents for vasoconstriction?

RESPONSE: We thank the reviewer for their question. Phenylephrine was chosen in this study as it is used widely in myograph studies assessing mesenteric arteries, which is due to the well-established presence of its receptors (alpha-1 adrenergic receptors) on these resistance vessels. However, we agree that assessment of other vasoconstrictors, particularly those relevant to preeclampsia, such as endothelin-1, which we know is elevated in the circulation of individuals with preeclampsia, would add value. Though this was out of the scope of this current study, we have acknowledged the benefits assessing multiple vasoconstrictors would add in our discussion.

Please see Page 12, Lines 338-340: “Assessing vascular responses with other dilating and constricting agonists may further validate the mechanisms by which esomeprazole acts on the vasculature.”  

  1. The statement “Using this model of L-NAME and obesity thus allowed us to insult the synthesis of nitric oxide at two points, substrate transport and enzyme activity” is too strong, from my opinion. The research does not provide enough evidences on altered NO signaling pathway.

RESPONSE: We thank the reviewer for their comment. We have edited this statement accordingly, to state our aim was to intervene with the nitric oxide pathway at two points.

Please see Page 11, Lines 324-327: “Using this model of L-NAME and obesity, we thus aimed to insult the synthesis of nitric oxide at two points, substrate transport and enzyme activity.”

  1. Are there any connections with the inflammatory status in control and obese mice treated with the esomeprazole?

RESPONSE: In this study, we assessed circulating levels of C-Reactive protein, an inflammatory marker. C-Reactive protein is elevated in the maternal circulation of individuals with preeclampsia, and in our L-NAME model. However, esomeprazole was unable to reduce circulating C-Reactive protein in both the lean and obese mice (Figure 3C and 8C, respectively).

Finally, the current article provides a substantial information on how esomeprazole impact preeclampsia and associated pathologies in normal and obese models. Article can be accepted after the minor revision.

Reviewer 2 Report

The authors assess the effect of esomeprazole on the maternal vasculature in normal (lean) and obese pregnant mice with impaired NO synthesis, which can be considered as a model of preeclampsia. 

Despite bringing mainly negative results it is a valuable study adding a new puzzle to the complex mechanisms of preeclampsia and the potential role of esomeprazole in the prevention of preeclampsia. The authors found that esomprezole enhanced ex-vivo acetylcholine-induced vacorelaxation. 

The manuscript is well written and self-explanatory. I have only few comments:

1. Figure 2 also shows some trend towards significance. What were the numeric results of the statistical test? I do not see any p-values presented?

2. Figure 3 - please explain the * and **

3. Please unify the terminology - normal vs. lean mice and maybe use one term throughout the manuscript - lean mice. 
